# Understanding Cancer Survivorship among Firefighters: A Mixed-Method Study

**DOI:** 10.3390/ijerph20010257

**Published:** 2022-12-24

**Authors:** Natasha Schaefer Solle, Amy Legros, Rachael Jackson, Apoorva Rangan, Cynthia Campos Beaver, Hannah Kling, Fatima Khan, Tulay Koru-Sengul, Frank J. Penedo, Alberto J. Caban-Martinez, Erin N. Kobetz

**Affiliations:** 1Departments of Medicine, Leonard M. Miller School of Medicine, University of Miami, Miami, FL 33136, USA; 2Public Health Sciences, Leonard M. Miller School of Medicine, University of Miami, Miami, FL 33136, USA; 3Sylvester Comprehensive Cancer Center, Leonard M. Miller School of Medicine, University of Miami, Miami, FL 33136, USA; 4Psychology, University of Miami, Coral Gables, FL 33146, USA

**Keywords:** survivorship, occupational health, cancer, mixed methods

## Abstract

Background: Firefighters are exposed to a unique set of carcinogens through their work environment that predispose them to several cancers, yet there is limited research related to cancer survivorship amongst this occupational group. Methods: A mixed-method approach was used to assess cancer survivorship amongst firefighters. Four focus groups and one in-depth interview were conducted with 29 active and retired firefighters who have been diagnosed with cancer to understand the experiences and challenges associated with cancer survivorship in the fire service and desired resources. Qualitative data were analyzed using Nvivo software. All participants completed the Functional Assessment of Cancer Therapy-General (FACT-G) survey to assess their quality of life. Results: The primary themes that emerged from the focus groups included managing health and well-being changes, navigating support systems in place, and accessing new resources. FACT-G scores (mean ± standard deviation) of the firefighter cancer survivor sample demonstrate relatively lower levels of emotional well-being (19.26 ± 4.67) and higher quality of life in the physical well-being (23.67 ± 5.08), social well-being (23.38 ± 4.16), and functional well-being (22.6 ± 4.966) domains. Conclusions: Firefighters requested curated resources, such as support groups and department training resources, supporting the need for more cancer survivorship resources specific to firefighters.

## 1. Introduction

Firefighters have unique occupational exposure to carcinogens [1] and associated increased risk of several cancers over the general population [2,3,4,5]. From January 2002 to March 2017, cancer has attributed to 61 percent of career firefighter line-of-duty deaths, according to data from the International Association of Fire Fighters [6]. The current survival rate for all cancers combined is 70% among white and 64% among black Americans [7], and it is known that the number of cancer survivors are increasing due to better screening and outcomes in oncology [8]. With cancer affecting the fire service in such a way, it can be estimated that there are many firefighters that are cancer survivors, defined as the period from cancer diagnosis to death [9].

Though it has been established that firefighters are exposed to unique carcinogens from their work environment, a lack of widespread reduction in cancer incidence and mortality amongst the U.S. firefighter workforce has been reported [10,11,12]. Our team previously reported that cancer is a concern for career firefighters at both early and late stages of their careers [13]. Cancer survivors are extensively analyzed in the literature; however, there is a dearth of analysis of firefighters who survived cancer. It is well-known that cancer survivors experience many changes in their mental and physical health, and that navigating these changes is improved with support from their medical team or family [14]. Additionally, it has been reported that cancer survivors have many unmet needs, with the highest number of unmet needs in activities of daily living domain, followed by psychological, information, psychosocial, and physical as the most frequently reported [15]. The highest number of individuals with unmet needs reported across multiple domains was in the post-treatment or survivorship phase [15], suggesting that there are unmet needs of firefighter cancer survivors. The purpose of this qualitative work is to characterize the management of mental and physical health, support accessed, and resources needed by firefighter cancer survivors based on focus group discussions.

## 2. Methods

### 2.1. Study Design, Participants, and Recruitment

A mixed-method approach was used to learn about the experiences of firefighter cancer survivors and their reported quality of life. A qualitative descriptive design (i.e., focus groups and interviews with a cross-sectional, demographic survey) was conducted with a total of four focus groups and one in-depth interview with 29 active and retired firefighters who identified themselves as cancer survivors. In addition, participants were asked to complete the Functional Assessment of Cancer Therapy-General (FACT-G) survey to assess their quality of life. Participants were considered eligible to participate if they were an active or retired firefighter with a current or prior cancer diagnosis. Focus group and interview sessions took place at local fire service administrative buildings and virtually via Zoom. Recruitment of participants was accomplished in collaboration with our established firefighter partners including department leadership, union leadership, and firefighter support organizations. Details regarding study enrollment were distributed through email Listservs, posted on social media, and presented by the research team at monthly calls hosted by firefighter organizations. 

### 2.2. Focus Group/Interview Script, Survey Measures, and Administration

A semistructured script was developed to gain an understanding of the experiences, challenges, and overall quality of life of firefighter cancer survivors. The interview guide was structured in 4 areas: work, health, and cancer diagnosis; challenges and outlook; effect of cancer on your career; and ideal workplace for cancer and return to work. All focus groups and interviews were audio-recorded and transcribed word for word. 

### 2.3. Data Analysis for Quantitative and Qualitative Data

Means and frequencies were calculated for sociodemographic information (Table 1). Quality of life was assessed using the FACT-G survey, version 4, which is a 27-item questionnaire that measures four quality-of-life domains: physical well-being (PWB), social well-being (SWB), emotional well-being (EWB), and functional well-being (FWB). Questions in all domains are measured using a 5-point Likert scale. Means of each subscale were calculated as directed in the FACT-G Scoring Guidelines [1]. Total score was computed by adding the subscales and could range between 0 (lower quality of life) to 108 (higher quality of life). Mean and SD of each subscale and total score were calculated (Table 2). Analysis for both descriptive and quality-of-life data was completed using SAS (SAS 9.3, SAS Institute, Inc., Cary, NC, USA) analytic software and Microsoft Excel (Version 16.42).

A constant comparison method was used to analyze the qualitative data. Themes were inductively developed, and open coding was completed, creating first line coding for each phrase of the transcript. Codes were compared to identify themes among the focus groups and interviews. Data saturation was reached by comparing patterns within the individual focus groups and comparatively with all groups. Data analysis was led by a qualitative research expert with a team of researchers to distinguish key topics that emerged from the focus groups. This research protocol was reviewed and approved by University Institutional Review Board (IRB# 20180723). 

## 3. Results

### 3.1. Quantitative Results

Our sample of 29 firefighter cancer survivors had a mean age of 53 ± 10.19 years, were mostly male (90%), white (100%), and non-Hispanic (86%). The majority of our sample were married (76%), had children (90%), and attended college for one to three years (79%). About 93% of the sample did not have more than one job, were not previous smokers (81%), and participated in vigorous physical activity for at least 30 min three times per week (63%). The sample of cancer survivors reported having a history of skin (nonmelanoma) cancer (19%), prostate cancer (15%), colon cancer (13%), and leukemia (9%). Nearly 68% participants reported that they were able to return to work after treatment. Of those who answered, 75% affirmed that their immediate supervisor understood their health situation/health problems and believed that employees with health problems are treated well within the organization. Eleven percent of our sample decided not to inform their coworkers of their cancer diagnosis. Most participants (89%) stated that they were not at all losing hope in the fight against their illness.

FACT-G scores (mean ± standard deviation) of our firefighter cancer survivor sample were: PWB 23.67 ± 5.08; SWB 23.38 ± 4.16; EWB 19.26 ± 4.67; FWB 22.6 ± 4.966; and total 87.87 ± 12.92. Each of these means are relatively high in their respective ranges, meaning our sample of firefighter cancer survivors have a relatively high quality of life. 

### 3.2. Qualitative Results 

Three main themes emerged from the data regarding the priorities of the interviewed firefighter cancer survivors: managing health and well-being, navigating existing support systems, and requesting new resources. The main themes were supported by nine overarching categories describing specific concerns (Figure 1).

#### 3.2.1. Managing Health and Well-Being

Focus group participants were asked about the aspects of their work that encouraged or discouraged them from engaging in healthy activities while navigating survivorship. They discussed mental health, particularly their attitude toward their cancer diagnosis, occupational stress, and the dismissive nature toward cancer risk in certain departments. In terms of physical health, they discussed long work hours and lack of sleep common among their profession, their inability to perform tasks due to their cancer, and changing their habits to improve their health. The final discussion point focused on their change in outlook on life since being diagnosed.

#### 3.2.2. Mental Health

Firefighters are generally more aware than the general public of the carcinogens they encounter in their environment. Several of our focus group participants discussed how this awareness affected their response and attitude toward their cancer diagnosis. Notably, some were not surprised and/or treated the situation in a matter-of-fact way. These participants stated that the higher-than-normal cancer rates among firefighters contributed to their unsurprised reaction. As one participant stated, 


*I really was expecting it, and I don’t say that from bravado, it’s just how do you not walk away from this job without getting cancer? I don’t know how you don’t, you know?*


Although some of the participants were not surprised, a few discussed the emotional toll they experienced after hearing their diagnosis. One participant reported, 


*I thought I was going to die… I was just like; all right, well we’ll just do what we can do until my time is up. I mean I literally—I cried probably I don’t know, every day until my surgery because I just didn’t know like what was what. And I’m not going to lie. I went into that surgery thinking I wasn’t going to wake up.*


Participants also discussed how occupational stress impacted their overall health and may have contributed to their cancer diagnosis, in addition to the environmental carcinogenic exposures. A female firefighter stated, 


*Not only is it all the, the soot and the carcinogens and all of the things we run into, it’s the stress of the job. And I believe that stress is such a huge factor in cancer… If I had a bad day I’d swear I’d go home and think like I just gave myself cancer. Like it’s so ridiculous, but I feel like it’s—I feel like it’s a bigger factor than anything.*


Participants also expressed increased anxiety over being rediagnosed with cancer. As one survivor said, 


*Because that psychological problem that you feel, is unreal. Even now, I mean, it’s been a long time; I still get my blood tested almost monthly. And as I’m getting close to the month when I’m gonna go get my blood checked, you have that anxiety, “Are they gonna find something? Is there anything gonna be there?”*


Denial was also a key emotion identified by the participants both in describing their feelings about their diagnosis and the overall culture of the fire service related to cancer risk. Many believe that they and their peers dismissed their cancer risk as firefighters. “You know what I said to the cancer insurance when I was 40? ‘What would I need that for?’” Another participant added, “they’re always pushing getting checked out. And to be 100 percent honest, I said, ‘I’m not doing that, won’t do it; not gonna do it’.” Participants also discussed the general denial that occurs at a department level when a firefighter in the department is diagnosed with cancer. “‘It’s not gonna happen here,’ right? It was an individual event, not a cultural event in the department. And that’s what has to change.”

#### 3.2.3. Physical Health

In addition to the mental toll of the fire service, participants also noted the physical stress of the job, including “sleep deprivation, stress levels being high, exposure to all these other terrible things” and the role it played in their diagnosis. One participant stated,


*I think part of my problem and why I got the cancer that I had was related to stress, from lack of sleep and it’s kind of a double-edged sword, I mean it’s—it plays through all these things and now I know that.*


All of the interviewed firefighters reported their work schedule to be one that consists of being on for a 24 h shift and being off for 48 h. Some of them prefer this work schedule because they are able to have days to themselves for leisure or other commitments. However, many of the participants agreed that the long work hours and late-night shifts did not promote a healthy lifestyle before or after their cancer diagnosis. One participant stated, 


*This job personally makes me feel the worst because I’m not sleeping… you know it’s that thing where you try to eat good but then it’s eight o’clock at night and you haven’t eaten and you’re on the road and you just grab whatever.*


Moreover, some participants discussed that the days they are not working cannot always be used for leisure and maintaining healthy behaviors. Participants stated that their days off were dedicated to recuperating from long shifts and trying to “get back to normal”. They expressed that maintaining healthy habits was difficult because they were “trying to live healthy even when they’re exhausted”.


*You know how it is. You get off in the morning and you can’t go right to sleep because you’re wound up. Anybody that works the night shift is like that, so… your rhythm is off… so it is a hazard of the job.*


Participants also described how they have chosen to deal with the long-term physical effects of cancer. Many reported exercising more to mitigate the fatigue. Others mentioned bringing food from home in order to keep themselves from eating unhealthy food at the station or at a nearby restaurant. “I changed how I eat, you know, I had to get rid of some weight. I did all that and I continue to do that because I wanna stay alive”.

Furthermore, some stated that they changed their sleeping habits because their cancer diagnosis made them tired, and they no longer wanted to sacrifice their sleep time. This need led some of them to seek light duty and avoid field duty.

#### 3.2.4. Changes in Outlook

As a result of being diagnosed with cancer, several participants stated that their outlook on life shifted, including treating others differently and being grateful for being alive.


*I don’t complain about the pancreatitis because I’m here. Everything, there’s just a much more bigger purpose for me now and it’s, it’s definitely—I hate, this is going to sound so crazy, but it did. The cancer made me a better person, and I’m almost thankful for it because I, I was. I was a jerk.*


Many also reported living more in the “here and now” and being more mindful of how they spent their time. They noted spending more time with family and taking more vacation time rather than working to accumulate money. As one firefighter said, “the outlook is: appreciate every day, because I was invincible until—until I had—[cancer]” Another participant agreed saying,


*I think I’ve been back for 16 shifts, and I’ve taken probably four weeks, four to five weeks off because I’ve had the personal time off and it’s my vacation time and I’m willing to go out there and just enjoy the time off with the family and live for today rather than worry about it.*


#### 3.2.5. Navigating Existing Support 

When asked questions related to the diagnosis and challenges of cancer, focus group participants stressed the importance and necessity of support to meet said challenges. Forms of support identified during the focus groups can be divided into three distinct subcategories: family support, peer support, and organizational support. 

#### 3.2.6. Family Support

Support from family members and friends were among the most frequently mentioned forms of support, with family support often being highlighted as the most important form of support during the diagnosis and treatment process. One participant noted “You know, I got my family around me, because in the end, I don’t care who you are, when you’ve got that, it’s you and your family, period”.

Though support from family and friends was noted, the degree to which individuals shared the diagnosis with their family vastly varied. Some participants reported having large support systems of friends or family, while other individuals shared the information with a limited number of individuals. For example, one participant remarked that


*I was talking to some, you know, like three close friends, you know, about what was goin’ on, what the treatment—in fact, I didn’t even tell anybody for a month um, that I was diagnosed with it except for my—for my father.*


Many participants emphasized the support provided by their spouses, noting how critical it was to have them by their side during the diagnosis and treatment. Some participants also highlighted that spouses played a major role in navigating the steps needed to address the diagnosis and provided key education during the treatment process.


*If I wouldn’t have had my wife, the wife that I had...she was my education; that’s what you need. I think when you get diagnosed, you need education.*


Additional remarks noted that impacts of the diagnosis were not limited to the individual itself but had profound impacts on the family. While survivors may have felt supported during the diagnosis and treatment process, some individuals noted that their family members needed additional support and resources. 


*So, they told me afterwards, but they said we were a wreck. Like we were already you know kind of planning for you know thinking there was going to be a funeral. So, it was, it was pretty dark. I tried to not be depressed because I was trying to keep [my wife] from getting like that.*


#### 3.2.7. Peer Support

Some participants were open about their diagnosis with their peers. As such, support from fellow peers within the service was noted and greatly appreciated for many participants. For example, one individual noted, “you do rely on the team concept of getting you through things”. The degree of support received varied amongst participants, with forms of support ranging from receiving emotional support to their peers becoming active in a bone marrow registry to fundraising efforts conducted by the department. Support was also provided during active fire responses. The theme of the brother/sisterhood of the fire service was emphasized by both retired and working firefighters. One retired firefighter said,


*I put [my diagnosis] out on the retiree site; plus, I put it out on my Facebook page because I grew up here since 1955. And I got a lot of people that, you know, called and concerned. People... I haven’t talked with since I left, and he called me; a lot of the firefighters called me.*


One participant remarked about the benefit of having support from firefighters from other entities in their personal lives, given the unique nature and danger of their profession:


*But I think the firefighters’ support is better because normal people can’t relate to a lot of the images, and talking, and thoughts, and stuff that—the way we relate to things. I mean, we look at death a lot different a lot of times, and we’ll joke about it; whereas somebody else is mortified about it when you start talking about it.*


While support was provided on an individual basis amongst peers, the degree of department support and local support was varied. For some individuals, department support was often limited to sick days or was completely missing. Importantly, desire for increased support from the department was noted across multiple focus groups. As one firefighter remarked, “But my department, other than letting me have my sick time... other than that, you know, I got nothing from them. And—and in a way I still don’t”.

The importance of having a strong social support network and open dialogue regarding the realities of cancer within the fire service was touched upon in many focus groups. Another sentiment that was highlighted across multiple focus groups was the importance of having an individual who had gone through the process of cancer diagnosis and treatment as another resource. Many survivors noted their desire to pay it forward to firefighters who may be diagnosed in the future. One retired cancer survivor noted how they adopted the role of being a liaison for cancer education in their department: 


*Having somebody [is] probably one of the most important things... when somebody gets cancer and our people already have gotten cancer, I’ve literally walked them through what they need to do. They have any questions; they can call me at any time of day, and I give them the correct information to move forward on.*


#### 3.2.8. Organizational Support

Many individuals were familiar with national organizations and academic institutions dedicated toward providing aid to firefighters and their families and cancer awareness and prevention training. One individual cited utilizing the resources provided by such entities as an alternate resource to resolve deficits caused by lack of local or department support. Individuals remarked how representatives from these organizations provided crucial information on how to choose appropriate healthcare providers, such as the necessary surgeons and oncologists, recovery process, and other vital information. Overall, a positive reaction was felt toward the resources provided by such entities:


*It helped me to make decisions along the way as we were going from network to network trying to figure out; okay, is this the doctor that I feel comfortable with? Is this the surgeon? So, I thought that at least that portion of it, the cancer support network, was something that was of great value at that time.*


#### 3.2.9. Needed Resources

Although focus group participants stated many areas of support, as discussed above, they also made requests for additional resources that would have proven helpful during their cancer diagnosis and treatment process. The main resources cited as most needed in the fire service for survivors include centralized cancer educational resources, designated firefighter cancer survivor support groups, and training for all members of the fire service regarding healthy practices.

#### 3.2.10. Centralized Cancer Educational Resources

The most frequent request among participants was for centralized educational resources about cancer. Many participants echoed the sentiment of becoming their own advocate during their treatment plan out of necessity to understand their prognosis, treatment options, side effects, and financial options. “So, you gotta be your own advocate. You gotta do your own research. You gotta ask a lot of questions”. Most participants stated that they used online resources to better understand aspects of their cancer diagnosis. However, for some, the use of online searching proved to be counterproductive:


*I tried to do the online looking. I know a lot of you guys had really good success. Me? It painted a doom-and-gloom. Because you’re researching and then you’re going down a rabbit hole.*


Additionally, one participant remarked that it was difficult to know if online sources regarding cancer treatments and outcomes were reliable, as many websites post “stuff that’s unregulated”. Participants agreed that finding information to understand their cancer diagnosis and treatment would be easier to do if there was an accessible, centralized educational resource which they knew was accurate and up-to-date.


*And there’s good websites, there’s good information and then there’s some that’s just really crap. Um, you know, to be able to have access to um, reputable good online stuff um, would be a help as well… and maybe have it vetted out.*


Participants also acknowledged that navigating financial resources and department benefits was difficult for some, even with the financial means to pay for cancer treatment and end of life preparations. A participant stated, “Having somebody in your house that has a good working knowledge as to how our insurance works, is invaluable”.

Some participants acknowledged their departments and national firefighter career organizations as sources of education that could be accessed.


*Our department did a good job putting together what we call a cancer FAQ sheet…coming up with something like a—like you guys are talking about um, a website where people can go to to get the correct information or somebody they can turn to uh, and talk to [someone at a national firefighter organization] is great, I used ‘em.*


#### 3.2.11. Firefighter Cancer Survivor Support Groups

In addition to a centralized cancer education resource, firefighter-specific cancer survivor support groups were requested by many of our participants as a place to learn from one another and have mutual understanding for their circumstances. Several participants cited our research focus groups as a good model for the type of group they would like to participate in; one stated “more support groups … yeah, fire support groups, like this group. Maybe not inclusive to the types of cancer”. 

Most participants agreed that these groups must be made up of solely firefighters or first responders to provide the most mutual understanding due to their similar team outlook on life.


*You have to have a drive, and firefighters, police, they have a drive; that’s why they’re here…And so, it’s the inside drive, but there’s also routine things, and you do rely on the team concept of getting you through things. So, that’s why the support group also should be there somewhere.*


Many participants cited the creation of these firefighter-specific cancer survivor support groups as a way to support future generations of firefighter cancer survivors, a version of paying it forward, as discussed in the Support section. While these support groups were requested by many participants, it must be noted that not all participants were interested. Those that were not interested cited privacy as their main reason for disinterest.

#### 3.2.12. Universal Firefighter Cancer Education

The increased use of SCBA, decontamination procedures, and education about health were discussed on many occasions. Many participants acknowledged the improvement in the fire service regarding health and safety becoming a priority. 


*Fire service leadership has come to understand the importance of health and safety. So, if it’s a health and safety issue that’s brought forward, it probably runs—runs to the top pretty quick. Some other issues are probably still slow to move, but health and safety I think is important.*


However, when asked how many participants and their families knew about cancer in the fire service, there was a mixed response. Responses indicating lack of awareness included, “Zip; nothing”, “Yeah, it was new. It was new”, and “I don’t think that’s something that we discussed in our family in depth at all”. 

To raise awareness, some survivors advocated for more cancer awareness education in the fire service by becoming cancer- or health-specific resource officers for their departments. One participant took it upon themselves to do volunteer cancer training on prevention and postcancer diagnosis actions.


*I put together uh, a training program for my own fire department as a volunteer… and that program... took off…until somebody actually comes into them and describes what things are like [they’ll say], you know, ‘ah cancer not a big deal. Not gonna happen to me’.*


## 4. Discussion

To our knowledge, this study is the first to comprehensively assess firefighter cancer survivors’ view of the culture of health in the fire service. Overall, participants stated that their respective departments discussed healthy practices, but occupational stress, long work hours, and lack of sleep made it difficult to maintain healthy habits. Notably, some participants believed that these stressors contributed to their development of cancer. In addition to work stress, some experienced anxiety over their ability to continue working and were fearful of being rediagnosed. These stressors may explain why some participants expressed frustration with leadership who do not acknowledge the elevated cancer risk in the fire service. The lack of acknowledgement by certain leadership may be explained by how firefighters perceive risk. DeJoy and colleagues found that firefighters are inclined to accept a number of risks to be perceived as aggressive and to avoid being seen as noncontributing; and newly trained firefighters, compared to experienced firefighters, were more likely to believe that effective firefighting could be conducted safely without taking on excessive risks [16]. This dichotomy contradicts the findings in our study because the average age of this study’s cohort was 53 years old, and several believed that their peers, particularly new firefighters, were not informed enough on the topic of cancer risk in firefighting. Therefore, cancer diagnosis may be a mediating factor in risk perception among firefighters; however, our sample size is not large enough to make this association. 

In addition, this is the first study to our knowledge which assesses both the types of support firefighter cancer survivors utilized and the support they advised research teams and/or national organizations to create in the future. Rates of death by suicide and mental health disorders, such as PTSD, are greater amongst firefighters than many other occupational groups [17,18]. Assessing for the extent, or lack thereof, of support in different domains is important because a study found that among professional firefighters, social support is critical for reducing occupational stress and suicidal ideation [19], both of which are elevated in this occupational group [20]. Given that our focus group participants remarked that peer support offset occupational stress, a similar association may be found among firefighter cancer survivors. However, a study with a larger sample size must be conducted to confirm this phenomenon. Furthermore, the highest prevalence of unmet needs among cancer survivors occur during the treatment phase [15]; therefore, these study results will help future researchers devise an effective tool for firefighters who are diagnosed with cancer. 

### 4.1. Implications for Cancer Survivors

Squiers and colleagues described how cancer patients often inquire and seek resources based on their cancer status, age, gender, and race [21]. Based on the results of this focus group, it is clear that firefighters desire resources that are curated for their occupational status. Our participants requested reliable cancer information similar to nonfirefighter cancer survivors in other literature [15,22] but also requested psychosocial support from like-minded peers and additional cancer-related occupational training. To meet the need for information and like-minded peer support, we propose the development of an application available to those in the fire service which offers validated cancer education resources and virtual or in-person firefighter cancer support groups. While these support groups were requested by many participants, it must be noted that not all participants were interested. Those that were not interested cited privacy as their main reason for disinterest. A second application, or an extension to the first application, could be created for fire service trainers and leadership to access training resources regarding cancer risk and prevention to not only follow national firefighter organization recommendations on cancer but also assuage our participants’ fears that younger firefighters do not know enough about the risk of cancer in the fire service. Not only would this training mirror gradual changes in occupational training currently occurring in the fire service, primarily with the advice from national firefighter organizations, but it may also improve access to federal provisions for employment-related needs for cancer survivors [23].

Furthermore, several participants believe that occupational stress contributed to their development of cancer. Therefore, our research team recommends that fire stations emphasize the importance of mental health and healthy coping mechanisms within their workplace. This can include providing firefighter resources surrounding mental health, fostering an environment that makes employees comfortable seeking help for occupational stress, and developing and/or advertising firefighter support groups. 

In addition to peer support and educational information, many participants were grateful for a local cancer center that worked with their fire station to streamline the process for firefighters to seek the care they needed, even if they chose to receive care from another center. Therefore, we recommend local cancer centers work with interested fire stations to create a working relationship.

### 4.2. Limitations 

Despite the challenges and barriers that the firefighters in our sample discussed, their FACT-G survey scores demonstrated that they had a high quality of life. This is likely due to the fact that most of the participants in our sample completed their cancer treatment. Therefore, many of our participants likely no longer felt physically limited, emotionally unstable, or consistently ill. These changes likely lead to high physical well-being, emotional well-being, and functional well-being scores, respectively. As some of our participants stated in the study, being diagnosed with cancer led them to lean more on their support system. This likely raised their social/family well-being score. A sample where many of the participants are not in remission would likely show lower FACT-G scores; therefore, our sample is not generalizable to firefighters in different stages of survivorship. Lastly, two of our participants did not complete the FACT-G survey, which could have skewed the data.

Furthermore, cancer type and severity of symptoms may affect survivorship perspectives. However, this likely did not affect our study because of the variety of cancer types represented. Subanalyses by cancer type were not performed due to our limited sample size. 

Our sample was closely representative of national firefighter demographics in terms of sex and race [24]. After both the Hispanic and non-Hispanic White races, Black and Asian are the second- and third-most represented races in the fire service, making up about 7% and 1% of the cohort, respectively. Unfortunately, non-White individuals did not apply to our study. The low number of non-White individuals in the fire service nationally and the convenience sampling in this study likely led to the lack of non-White minorities in our sample. This limitation may exclude race- or ethnicity-related factors in cancer survivorship. Studies should be conducted to investigate if minorities in the fire service have different perspectives on cancer survivorship and different requests for cancer resources. 

Additionally, recruitment through local firefighter social media outlets and Florida firefighter organizations made the sample a convenience sample. For this reason, many participants were Florida residents. The concentration of participants in one state and the small sample size of 29 means that our results are not generalizable to all firefighters across the US. However, the main takeaways from this paper remain important in providing insight to the survivorship needs of firefighters.

## 5. Conclusions

This mixed-methods study characterized the management of health changes, support access, and unmet needs of firefighters who survived cancer. While our participants had high quality-of-life scores, they requested curated resources, such as support groups and department training resources. This study supports the need for cancer survivorship resources specific to firefighters.

## Figures and Tables

**Figure 1 ijerph-20-00257-f001:**
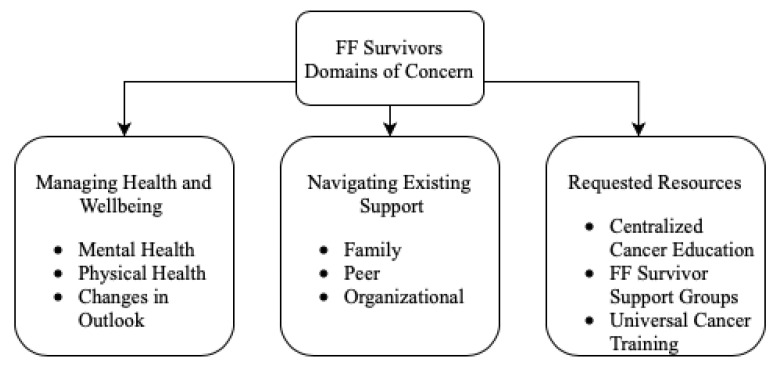
Themes and subthemes evidenced in focus groups.

**Table 1 ijerph-20-00257-t001:** Socio-demographic and cancer characteristics of firefighter cancer survivors participating in the firefighter cancer survivorship focus groups and survey.

Characteristics	Total Sample *n* = 29 *
	Mean (SD)
Age (years)	53 (10.19)
	N (%)
Gender	
Male	26 (89.66)
Female	3 (10.34)
Ethnicity	
Hispanic	4 (13.79)
Non-Hispanic	25 (86.21)
Race	
White	29 (100)
Marital Status	
Married	22 (75.86)
Divorced	1 (3.45)
Widowed	1 (3.45)
Separated	2 (6.90)
Never married	2 (6.90)
Member of an unmarried couple	1 (3.45)
Education Level	
College 1–3 years	23 (79.31)
College 4 or more years	6 (20.69)
Children	
Yes	26 (89.66)
No	3 (10.34)
Currently has more than 1 job	
Yes	2 (6.90)
No	27 (93.10)
Currently a tobacco smoker	
Yes	2 (7.41)
No	25 (92.59)
Past tobacco smoker	
Yes	5 (18.52)
No	22 (81.48)
Participates in vigorous PA for 30 min 3 times/week	
Yes	17 (62.96)
No	10 (37.04)
Self-reported type of cancer	
Blood	1 (3.13)
Breast	1 (3.13)
Colon	4 (12.50)
Throat/nose	2 (6.25)
Leukemia	3 (9.38)
Liver	1 (3.13)
Lung	2 (6.25)
Lymphoma/Hodgkin’s disease	1 (3.13)
Melanoma	1 (3.13)
Pancreas (pancreatic)	1 (3.13)
Prostate	5 (15.63)
Skin (nonmelanoma)	6 (18.75)
Testis	1 (3.13)
Other	3 (9.38)
Number of different cancers diagnosed	
1	21 (80.76)
2	5 (19.23)
Currently has other cancer	
Yes	2 (50)
No	2 (50)

* Differences in subtotal population sample due to nonresponse or missing data.

**Table 2 ijerph-20-00257-t002:** Results from FACT-G survey among a sample of firefighter cancer survivors (*n* = 27).

Subscale	Mean Score (SD)
Physical Well-Being (PWB) ^a^	23.67 (5.08)
Social/Family Well-Being (SWB) ^b^	23.38 (4.16)
Emotional Well-Being (EWB) ^c^	19.26 (4.67)
Functional Well-Being (FWB) ^d^	22.6 (4.966)
Total FACT-G Score ^e^	87.87 (12.92)

^a^ PWB scores may range from 0–28; ^b^ SWB scores may range from 0–28; ^c^ EWB scores may range from 0–24; ^d^ FWB scores range from 0–28; ^e^ FACT-G total may range from 0–108.

## Data Availability

The data presented in this study are available on request from the corresponding author. The data are not publicly available due to participant confidentiality.

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
