# Peer review of "Understanding Cancer Survivorship among Firefighters: A Mixed-Method Study"

_ijerph, 2022, doi:10.3390/ijerph20010257_

Round 1
Reviewer 1 Report
This is a well written manuscript describing a mixed method study on cancer survivorship of firefighters.
It discusses an important problem facing this occupational group worldwide. This is an important contribution to firefighters in different parts of the world and has a relevance to a wider audience and merits publication.
One point which could be considered by authors is discussion around the difference between qualitative findings (focus groups and interviews) vs FACT G results. FACT G scores show very high quality of life which doesn't seem to be much supported with the findings from qualitative results. Authors may like to add some comments about this somewhere in the discussion.
Reviewer 2 Report
Comments to the authors
Reviewer: General comments
The authors conducted a qualitative study that aimed to characterize social, mental, and physical determinants of firefighters’ survivorship after cancer. The authors report 4 primary themes that emerged from the focus group and overall, high scores in terms of Well-being. In general, this is a well conducted study that contributes to better understand cancer survivorship determinants in an occupational group that is now recognized as carcinogen.
Major comments
- Please, replace the first reference (‘Painting: firefighting, and shiftwork’. IARC Monogr Eval Carcinog Risks Hum.2010) with the last IARC monography about firefighters entitled ‘Occupational exposure as a firefighter’ (2022).
- Cancer survivorship determinants may be strongly influenced by the cancer type and its severity. Did the authors conduct sub-analyses that consider one of these parameters? If not, the authors should discuss the reasons why they did not.
- Could the authors explain the reason why there were only white subjects in their study sample and it potential impact in the results that they report here?
Minor comments
- Did the authors have access to any objective source to confirm the subjects self-reported anatomical localisation of cancer?
- Page 3, page 4: If possible, the authors should precise the reasons why 2 subjects were missing in the FACT -G survey.
- For greater clarity, the authors should enclose firefighters’ statements in quotation marks.
